# Ginsenosides and Polysaccharides from Ginseng Co-Fermented with Multi-Enzyme-Coupling Probiotics Improve In Vivo Immunomodulatory Effects

**DOI:** 10.3390/nu15112434

**Published:** 2023-05-23

**Authors:** Shaowei Bai, Guangyun Zhang, Yaqin Han, Jianwei Ma, Bing Bai, Jingjie Gao, Zuoming Zhang

**Affiliations:** Key Laboratory for Molecular Enzymology & Engineering of the Ministry of Education, School of Life Sciences, Jilin University, Changchun 130012, China; baisw21@mails.jlu.edu.cn (S.B.); zhanggy1@163.com (G.Z.); h95631817@163.com (Y.H.); jwma20@mails.jlu.edu.cn (J.M.); baibing20@mails.jlu.edu.cn (B.B.); gaojj21@mails.jlu.edu.cn (J.G.)

**Keywords:** ginseng, enzyme, probiotics, immunosuppression, intestinal flora

## Abstract

The active components of ginseng, such as ginsenosides and polysaccharides, have high therapeutic value in treating cancer, decreasing obesity, and enhancing immunity. However, simple primary ginseng treatment cannot maximize this medicinal potential. Therefore, in this study, *Panax ginseng* was co-fermented with multi-enzyme-coupling probiotics to obtain a fermentation broth with higher levels of ginsenosides, polysaccharides, and probiotics. When compared to other treatment methods for cyclophosphamide-induced immunosuppression in mice, the results reveal that the *P. ginseng* fermentation broth treated with multi-enzyme-coupling probiotics could significantly improve the immune function of immunosuppressive mice and restore intestinal flora stability. Overall, this processing method will provide a novel strategy for promoting the application of ginseng and the relief of immunosuppression.

## 1. Introduction

*Panax ginseng*, a perennial herb from stolon, is mainly distributed in northeastern China, with a small distribution in Japan and Korea and poses a wide range of pharmacological activities [1,2]. Ginseng contains various active components, such as ginsenoside, polysaccharides, peptides, amino acids, etc. [3], exhibiting different activities like anti-diabetic, anti-inflammation, and anti-oxidation [4]. It is reported that ginseng exerts a significant effect during the adjuvant treatment of many diseases, such as hypertension, as well as Alzheimer’s and Parkinson’s disease [5]. Ginsenosides are regarded as the active component in ginseng [6,7] and are classified into dammarane-type ginsenosides and oleanane-type ginsenosides according to the difference in the position and quantity of sugar moiety in the glycosides. The dammarane-type ginsenosides are subdivided into protopanaxadiol (PPD) and protopanaxatriol (PPT), while the oleanane-type ginsenosides are mainly oleanolic acids. According to the content differences in the ginseng, ginsenosides can be classified into major ginsenosides and minor ginsenosides. Early studies have found that the medicinal value of minor ginsenosides is better than that of major ginsenosides [8,9]. Nevertheless, the content of ginsenosides extracted from unprocessed ginseng cannot meet medical needs, as most belong to the major ginsenosides with low medicinal values. Therefore, transforming the major ginsenosides into minor ginsenosides is of great significance. At present, the common methods used for ginsenoside transformation include physical, chemical, and biological methods [9]. The physical method requires harsh conditions, with more byproducts, and is not conducive to purification. The chemical method has many disadvantages, such as selectivity, isomerization, and environmental pollution. In contrast, biotransformation utilizes organisms or enzymes as catalysts to achieve ginsenoside transformation with high selectivity, mild conditions, fewer byproducts, and less ecological impact [10,11]. Ginseng polysaccharides (GPs) form an essential structural component in ginseng cells and are released by cell wall deconstruction. GPs contain different monosaccharides, such as arabinose (Ara), rhamnose (Rha), glucose (Glu), galacturonic acid (GalA), and galactose (Gal), due to their different sources and purification processes [12]. It is reported that GPs can stimulate immune cells and promote the release of immune factors, thus regulating immune function [13,14,15]. Briefly, ginseng can be used as a dietary supplement to regulate the function of the immune system, which might be closely related to the active components in ginseng, namely minor ginsenosides and GPs.

The gastrointestinal tract is the largest immune organ in the body and plays a significant role in regulating immune function [16]. Meanwhile, the homeostasis of intestinal flora is conducive to the normal development of intestinal immune function [17]. Probiotics are active micro-organisms that can promote the ecological balance of intestinal flora and are beneficial to host health [18]. Currently, many probiotics are used to treat gastrointestinal discomfort and improve the function of the immune system [19,20]. However, the mechanism underlying the improvement of immune function by probiotics has not been fully elucidated yet.

Therefore, in this study, ginseng was treated with a multi-enzyme-coupling probiotic to improve the transformation and release of the active components of ginseng. The method was optimized by an orthogonal test. The results indicate that the obtained ginseng fermentation broth could effectively regulate the immune function of immunosuppressed mice and help restore their intestinal flora homeostasis. This study provides a novel strategy for ginseng processing and lays a theoretical foundation for the development of new ginseng products to improve immune function.

## 2. Materials and Methods

### 2.1. Micro-Organisms and Materials

The ginseng used in this experiment was 4–5 years old fresh ginseng and was cultivated in Baishan, Jilin Province, China. The ginseng was cleaned and sterilized using high pressure (121 °C; 20 min), and was then packed in a vacuum and frozen at −20 °C. The sources of the fermentation micro-organisms are shown in Appendix A.

### 2.2. Pretreatment of Ginseng

The ginseng was pretreated by the homogenization method. Briefly, the frozen ginseng was defrosted and broken into small pieces, and the distilled water was then added at a material-to-liquid ratio of 1:4. The crude ginseng pulp was obtained by stirring with a blender, and then the crude ginseng pulp was stirred with a high-speed disperser at 10,000 rpm for 10 min. The final product was sterilized at 121 °C for 20 min after passing through a 40-mesh sieve and cooled to room temperature.

### 2.3. Selection of Strains

The pretreated ginseng pulp was used as a medium for fermentation experiments using four micro-organisms, namely *Streptococcus thermophilus*, *Lactobacillus plantarum*, *Lactobacillus rhamnosus*, and *Bacillus amyloliquefaciens*. The ginsenosides in the fermented ginseng pulp were analyzed using thin-layer chromatography (TLC). The strains were selected for ginseng fermentation by evaluating the ginsenosides transformation efficiency of the four micro-organisms.

### 2.4. Selection of Enzyme Agents

According to the structure of the plant cell wall, 16 kinds of enzymes were used to treat the ginseng, including cellulase (1), pectinase (2), amylase (4), protease (4), rhamnosidase (3), glucoamylase (1), and glucosidase (1). These enzymes were added to the pretreated ginseng pulp with a 1% enzyme content (pH 6.4; 37 °C). By analyzing the total soluble sugar (the phenol-sulfuric acid method) [21], reducing sugar (DNS method) [22], ginsenoside (HPLC), and protein contents (the Bradford method) [23] after enzyme treatment, three kinds of enzymes that effectively promoted the dissolution and transformation of the active components were selected for subsequent ginseng fermentation.

### 2.5. TLC and HPLC Analysis Method

TLC was used for the qualitative identification of the ginsenosides, using chloroform-methanol-n-butanol-water (13:10:10:8, *v*/*v*/*v*/*v*) as a developing solvent. The visualization reagents were sulfuric acid-methanol. The sample (1 mL) added with n-butanol (300 μL) was mixed and centrifuged, and the upper layer was taken for TLC.

The ginsenosides were quantitatively analyzed via HPLC. The specific method was as follows: the ginsenosides Rg3, F2, F1, Rd, Rg2, Rb2, Rc, Rb1, Rf, Re, and Rg1 were dissolved in chromatographic-grade methanol to prepare the standard solution with the concentrations of 0.4 mg/mL, 0.8 mg/mL, 1.2 mg/mL, 1.6 mg/mL, and 2.0 mg/mL, respectively, and then filtered via a 0.22 μm filter membrane and then reserved at low temperature. A gradient solvent (acetonitrile/H_2_O: 23% to 85%; t_R_: 0 to 61 min; acetonitrile/H_2_O: 85% to 23%; t_R_: 61 to 80 min) and a flow rate of 0.5 mL/min were used for HPLC, with a detection wavelength of 203 nm.

### 2.6. Bradford Analysis Method

The protein content of ginseng pulp was determined by the Bradford method using a protein assay kit (Beyoyime P0006). Briefly, the BSA protein standard solution (5 mg/mL) was diluted into a gradient solution with concentrations of 0, 0.125, 0.25, 0.5, 0.75, 1, and 1.5 mg/mL, respectively. The above dilutions (5 μL) were added to the sample wells, followed by the addition of the G250 staining solution (250 μL) to each well, and the absorbance was measured at 595 nm to plot the standard curve. The supernatant of the ginseng pulp was centrifuged (10,000 rpm; 5 min), and the protein concentration was calculated according to the standard curve after the reaction under the same conditions.

### 2.7. Orthogonal Test and Validation Tests

Based on the single-factor experiments and L9(3^3^) orthogonal table, the colony-forming units of *Lactobacillus rhamnosus* and *Bacillus amyloliquefaciens* (CFU/mL) were used as the evaluation index. The amount of selected enzyme addition was evaluated, and the amount of multi-enzyme addition conducive to the growth and propagation of probiotics was optimized. *Lactobacillus rhamnosus* (MRS medium) and *Bacillus amyloliquefaciens* (NA medium) were cultured overnight at 37 °C and 150 rpm, respectively. The pretreated ginseng pulp was added to the selected enzymes based on the orthogonal test results and treated at pH 6.4 for 2 h. Then, the buffer (2 g CaCO_3_) was added to the enzymatic hydrolysate to neutralize the acidity. Meanwhile, *Lactobacillus rhamnosus* (2%) and *Bacillus amyloliquefaciens* (2%) were inoculated into ginseng pulp (pH 6.4), respectively. Finally, the samples were collected at 37 °C and 180 rpm for 1 h and 24 h, respectively, and counted using the hanging drop method.

### 2.8. Co-Fermentation using Multi-Enzyme-Coupling Probiotics

The selected multi-enzyme combinations and probiotics were used for the co-fermentation of the ginseng. During fermentation, samples were taken regularly to determine the changes in the bacterial count, total soluble sugar, and reducing sugar contents. Finally, the number of viable bacteria in the final fermentation product was determined.

### 2.9. Animal Studies

The female BALB/c mice (18–22 g) from Liaoning Changsheng Biotechnology Co., Ltd. were raised in a sterile environment at a temperature of 25 ± 2 °C and were light-controlled with a 12 h light/dark cycle. The experimental procedures were approved by the Ethics Committee for the welfare of Jilin University Laboratory Animals. The experimental animals were randomly assigned into nine groups, as shown in Appendix A (NC: Normal Control; MC: Model Control; GC: Ginseng Control; LE: *Lactobacillus rhamnosus* Enzyme; LF: *Lactobacillus rhamnosus* Fermentation; BE: *Bacillus amyloliquefaciens* Enzyme; BF: *Bacillus amyloliquefaciens* Fermentation; LF&BF: *Lactobacillus rhamnosus* and *Bacillus amyloliquefaciens* Fermentation; PC: Positive Control). The experiment was conducted after one week of the adaptive period, and the immunosuppression model of the mice was established using cyclophosphamide (CTX) [24]. At the end of the experiment, the animals’ feces were collected via the stress defecation method and stored in an EP tube at −80 °C. The mice were killed, and their spleens were weighed and stored in 4% paraformaldehyde and −80 °C, respectively. The jejunum tissue of the mice was collected and fixed with 4% paraformaldehyde for morphological observation.

### 2.10. Evaluation of the Immunomodulatory Effect

The spleen index of the mice was calculated according to the following formula: spleen index (mg/g) = mean spleen weight (mg)/mean body weight (g) [25]. The proliferation of splenic lymphocyte experiments was carried out after the mice were killed. The spleen of the mice was crushed and sieved through a 200-mesh screen to prepare the spleen cell suspension. The cell suspension was washed thrice with phosphate buffer, suspended in RPMI1640 medium, and then the cell concentration was adjusted to 5 × 10^6^ mL^−1^. The prepared spleen cell suspension was added to a 96-well plate with 200 μL in each well. Except for the control group, 5 mg/mL concanavalin was added to the other groups and cultured at 37 °C for 48 h. Finally, each group was incubated with 5 mg/mL MTT for 4 h. The supernatant was collected after centrifugation at 1000× *g* for 10 min, and 150 μL of dimethyl sulfoxide was then added. The absorbance at 570 nm was measured after incubation in the dark for 15 min, and the proliferation activity of the splenic lymphocytes was calculated. The jejunum and spleen tissues were used for pathological observation, following specific procedures, such as fixation, dehydration, embedding, sectioning, and staining. The intestinal injury was observed at 100 times magnification, and the spleen injury was observed at 200 times magnification.

### 2.11. Analysis of Intestinal Flora

The fecal flora of the experimental animals was examined using 16S rDNA gene sequencing entrusted to Sangon Biotech (Shanghai, China) Co., Ltd. After the pretreatment of the stool sample, the DNA was extracted from the stool using the DNA extraction kit (OMEGA), and the DNA was initially examined according to the location and quantity of the nucleic acid electrophoresis bands. The precise quantification of the genomic DNA was conducted using the DNA detection kit (Qubit). The PCR primer sequences were CCTACGGGNGGCWGCAG and GACTACHVGGGTATCTAATCC, respectively. After two rounds of PCR amplification, the size of the DNA library was detected by 2% agarose gel electrophoresis, and the concentration of the DNA library was determined via a quantitative fluorescence instrument (Qubit). After the data obtained by Illumina Miseq^TM^/Hiseq^TM^ (San Diego, CA, USA) were transformed into the original sequencing sequence, the community composition and phylogeny of the micro-organisms in the samples were obtained via correlation analysis.

### 2.12. Data Processing

Data were analyzed using Origin8 (Lake Oswego, OR, USA) and GraphPad Prism 8 (San Diego, CA, USA) and are expressed as mean ± SD. The statistical analysis and analysis of variance were performed using IBM SPSS Statistics 23 (*: *p* < 0.05; **: *p* < 0.01).

## 3. Results and Discussion

### 3.1. Selection of Strain and Enzyme Agents

Ginsenoside is one of the active components of ginseng, and the medicinal value of minor ginsenoside is better than that of major ginsenoside. Therefore, the transformation of major ginsenoside into minor ginsenoside represents a potential way to improve the medicinal value of ginseng. As depicted in Figure 1A, the ginsenosides in the fermented ginseng pulp were analyzed via TLC after 24 h of single fermentation with four strains (*Streptococcus thermophilus*, *Lactobacillus plantarum*, *Bacillus amyloliquefaciens*, and *Lactobacillus rhamnosus*) to screen the suitable fermentation strain. When compared to the control group, the ginsenoside showed no significant transformation in the ginseng fermented with four micro-organisms. However, *Lactobacillus rhamnosus* and *Bacillus amyloliquefaciens* as probiotics could improve immunity and regulate gut microbiota homeostasis [26,27,28]. Additionally, *Lactobacillus rhamnosus* and *Bacillus amyloliquefaciens* could inhibit the multiplication of pathogenic bacteria and toxin production in the fermented products, thereby improving the quality of the fermented products. More importantly, rhamnosidase and amylase, which are produced by *Lactobacillus rhamnosus* and *Bacillus amyloliquefaciens*, might promote the degradation of ginseng granules and ginsenosides transformation. Therefore, *Bacillus amyloliquefaciens* and *Lactobacillus rhamnosus* were selected as the experimental strains for co-fermentation with the multi-enzyme-coupling probiotics.

After the enzymatic transformation of the ginseng pulp using 12 types of enzymes, the supernatant was collected via centrifugation to screen out the beneficial enzymes that promote ginseng co-fermentation. The total soluble sugar and reducing sugar contents in the supernatant were determined, and the results are depicted in Figure 1B. When compared to the control group, the total soluble sugar content in the supernatant of the ginseng pulp treated with 12 enzymes significantly increased, indicating that these enzymes could promote the release of soluble sugars. Of these, the catalytic effect of cellulase was the most significant, with a soluble total sugar content increasing from 43.20 mg/mL to 71.00 mg/mL. Meanwhile, the polysaccharide content in the supernatant treated with cellulase was found to be up to 43.37 mg/mL. These sugars, as good sources of carbon, can be used by probiotics in the subsequent fermentation process. Studies have shown that ginseng polysaccharides pose various physiological functions, especially in immunity and anti-tumor activity. Therefore, cellulase was selected as one of the enzymes to transform ginseng.

Later, the ginsenosides in the ginseng pulp treated with 16 enzymes were analyzed via TLC, and the results are depicted in Figure 2A–E. When compared to the control group, the ginsenosides in the ginseng pulp had been significantly transformed, and the contents of the minor ginsenosides (such as Rd, Rg2, F2, and Rg3) increased after treatment with cellulase, compound pectinase X, β-glucosidase, glycosylase, pectinase (solid), and rhamnosidase KDNO_3_. The ginseng pulp converted by these enzymes was then extracted using the alcohol extraction method, and the ginsenosides were quantified accurately using HPLC (Figure 2F). Of these, the total ginsenoside contents had increased the most after being treated with the compound pectinase X. When compared to the control group, the contents of the minor ginsenosides F2 and Rg3 in the compound pectinase X-treated group increased by 12-fold and six-fold, respectively. Since the compound pectinase X significantly increased the total ginsenosides and minor ginsenosides, pectinase X was selected as the second enzyme to transform the ginseng.

The soluble protein in the ginseng root accounts for 20% of the total soluble nitrogen, containing several amino acids and peptides. The protein content in the supernatant of the ginseng pulp treated with four proteases was determined by the Bradford method, and the results are depicted in Figure 3A. When compared to the control group (0.370 mg/mL), the flavor protease treatment group had the highest protein content (0.593 mg/mL), showing that the enzyme could degrade the ginseng root more fully and partially promote the release of ginsenoside and polysaccharide. Therefore, flavor protease was selected as the third enzyme to transform the ginseng.

### 3.2. Orthogonal Experiment

The ginseng pulp was treated with a single enzyme using flavor protease, cellulase, and compound pectinase X, respectively. The optimum addition amount of a single enzyme was determined by measuring the protein and total soluble sugar concentration in the supernatant, and the results are depicted in Figure 3B–D. The protein content in the ginseng pulp supernatant reached the highest level when the flavoring protease content was 0.15%. Meanwhile, the total soluble sugar in the ginseng pulp reached the highest level when the addition of cellulase and compound pectinase X was 0.20% and 0.6%, respectively. Therefore, the additions of 0.20%, 0.6%, and 0.15% of cellulase, compound pectinase X, and flavor protease were selected for the orthogonal test, respectively.

Based on the above single-factor test results, the orthogonal test was designed to obtain the optimal multi-enzyme concentrations (Appendix A). In the ginseng fermentation treated with multi-enzyme-coupling *Lactobacillus rhamnosus*, the orthogonal analysis of the results of L_9_(3)^3^ is shown in Table 1 and Appendix A. The influence (R) of additive content factors on the colony number was R_A_ > R_B_ > R_C_, and the optimal formula was A_2_B_1_C_2_, i.e., cellulase (0.20%), compound pectinase X (0.40%) and flavor protease (0.15%). Under these conditions, the colony number of *Lactobacillus rhamnosus* was 1.1 × 10^8^ CFU/mL.

Similarly, for the ginseng fermentation treated with multi-enzyme-coupling *Bacillus amyloliquefaciens*, the orthogonal analysis of the results of L_9_(3)^3^ is shown in Table 2 and Appendix A. The influence (R) of additive content factors on the colony number was R_B_ > R_C_ > R_A_, and the optimal formula was A_2_B_2_C_1_, i.e., cellulase (0.20%), compound pectinase X (0.60%), and flavor protease (0.10%). Three parallel experiments were carried out to verify the addition formulas of the multi-enzyme selected by the orthogonal test; the colony number of *Lactobacillus rhamnosus* and *Bacillus amyloliquefaciens* was 3.4 × 10^8^ CFU/mL and 2.4 × 10^8^ CFU/mL, respectively, which is better than the results of the other formulas. Therefore, A_2_B_1_C_2_ and A_2_B_2_C_1_ were the optimal formulas to promote the fermentation of *Lactobacillus rhamnosus* and *Bacillus amyloliquefaciens*.

### 3.3. Ginseng Co-Fermentation using Multi-Enzyme-Coupling Probiotics

Before co-fermentation, the reaction time catalyzed by the enzyme was determined, and the results are depicted in Figure 4A. The total soluble sugar content of A_2_B_1_C_2_ reached the highest at 2 h, accounting for 53.88 mg/mL. The total soluble sugar content of A_2_B_2_C_1_ was 49.60 mg/mL at 1 h. Therefore, for A_2_B_1_C_2_, *Lactobacillus rhamnosus* was inoculated for fermentation after 2 h, while for A_2_B_2_C_1_, *Bacillus amyloliquefaciens* was inoculated for fermentation after 1 h. Based on the above results, the change in the number of *Lactobacillus rhamnosus* during co-fermentation was counted by using the hanging drop method, and the results are depicted in Figure 4B. The colony number reached the highest level at 5 h (8.8 lgCFU/mL). The total soluble sugar content showed a decreasing trend, and the reducing sugar content was stable after increasing gradually (Figure 4C,D). Therefore, 5 h was selected as the endpoint for co-fermentation with multi-enzyme-coupling *Lactobacillus rhamnosus*. Similarly, the cell number reached the highest level at 9 h (9.2 lgCFU/mL) for co-fermentation with multi-enzyme-coupling *Bacillus amyloliquefaciens*. The soluble total sugar content reached its highest at 8 h (50.43 mg/mL), and the reducing sugar content gradually increased and finally remained stable (Figure 4C,D). Therefore, 8 h was selected as the endpoint for co-fermentation with multi-enzyme-coupling *Bacillus amyloliquefaciens*. The ginseng freeze-dried powder (co-fermentation with multi-enzyme-coupling *Lactobacillus rhamnosus*) was diluted 10^6^ times with normal saline, and the ginseng freeze-dried powder (co-fermentation with multi-enzyme-coupling *Bacillus amyloliquefaciens*) was diluted 10^7^ times with normal saline. The viable bacteria count for the freeze-dried ginseng powder is shown in Appendix A. The viable bacteria counts for the two fermented ginseng samples after lyophilization was above 10^10^ CFU/g, which could be used for subsequent animal experiments.

### 3.4. Immunomodulatory Effects In Vivo

Ginseng is frequently prescribed as a tonic in clinical settings. Its potent ingredients, such as ginsenosides and ginseng polysaccharides, can effectively improve immune suppression by promoting the weight of immune organs and the proliferation of splenic lymphocytes. As a result, we anticipated that co-fermentation using multi-enzyme-coupling probiotics would enhance ginseng’s immunomodulatory effects while also allowing us to identify the most effective ginseng treatment formula. The schedule for the animal experiments is depicted in Figure 5A. From 0 d to 3 d, the mice in NC were intraperitoneally injected with 0.2 mL saline, while the mice in other groups were intraperitoneally injected with 80 mg/kg/d CTX. From 4 d to 10 d, each group was treated according to the design shown in the experimental schedule. Various ginseng treatment groups (300 mg/kg/d) and the positive control (levamisole hydrochloride, 40 mg/kg/d) were administered to model the immunosuppressed mice. After the experiment, the feces and tissue organs were taken and stored. The spleen index analysis results are depicted in Figure 5B. The spleen index of the mice in the model group decreased by approximately 21.1% when compared to the control group, indicating that cyclophosphamide successfully induced the immunosuppression model in the mice. The spleen index of the LF group (co-fermentation with multi-enzyme-coupling *Lactobacillus rhamnosus*) significantly increased. In contrast, the effect of the BF group (co-fermentation with multi-enzyme-coupling *Bacillus amyloliquefaciens*) was poor. The difference in the effect of the two probiotic fermentation groups may be due to their different intestinal colonization ability. *Lactobacillus rhamnosus* are more suited to living in the anaerobic and acidic intestinal environment than *Bacillus amyloliquefaciens*. A splenic lymphocyte proliferation assay was performed, and the results are depicted in Figure 5C. When compared to the model group, the LF group showed significant splenic lymphocyte proliferation activity, indicating that the co-fermentation of ginseng with multi-enzyme-coupling *Lactobacillus rhamnosus* might exert immunomodulatory effects by promoting lymphocyte proliferation and attenuating CTX-induced immunosuppression.

The spleen is the largest immune organ. H&E staining was performed on the spleen tissues to evaluate the immunomodulatory effect of the different treatment groups, and the results are depicted in Figure 5D. The LF group had distinct red-pulp and white-pulp structures in the spleen, with fewer lymphocytes, indicating that ginseng co-fermented by the multi-enzyme-coupling *Lactobacillus rhamnosus* could partially restore the immune function of the mice. In terms of the maintenance of intestinal immunity, a histopathological evaluation of the small intestine revealed that the LF group could reduce the intestinal damage caused by immune suppression, such as elongated villi and a more normal intestinal wall (Figure 5E). Hong et al. used an enzyme-assisted extraction method to isolate functional polysaccharides from Korean ginseng and found that the functional polysaccharides obtained by this method represent a potential immune-stimulatory agent [29]. In contrast, our study utilized co-fermentation with a multi-enzyme-coupling probiotics method to treat ginseng, which improved the immunomodulatory activity of ginseng in many aspects and more fully reflected the medicinal value of ginseng.

Probiotics could maintain host intestinal flora homeostasis and participate in body immunity. Thus, the gut flora of the mice in the different treatment groups was analyzed. The microbial community structure was analyzed via principal component analysis (PCA) at the OUT level. As depicted in Figure 6A, the MC group was far from the NC group, indicating that the intestinal flora composition of the MC group showed a greater difference compared to the NC group. The LF group was closer to the NC group, indicating that the intestinal flora composition was similar. The principal co-ordinates analysis (PCoA) could be used to analyze the differences between the individuals or groups, as depicted in Figure 6B. The overlap between the LF and PC groups indicated that the community structures of the multi-enzyme-coupling *Lactobacillus rhamnosus* group and the positive group were similar. Then, the community structure of the different samples at the phylum level was compared, as depicted in Figure 6C. The community structure of the NC group mainly included *Firmicutes* (48.321%), *Bacteroidetes* (46.165%), and *Proteobacteria* (3.447%). In the MC group, the abundance of *Bacteroidetes* decreased, and the abundance of *Firmicutes* and *Proteobacteria* increased, indicating that the immune function of the mice was affected and the decreased resistance to pathogenic bacteria led to an increased abundance of pathogenic bacteria in the intestinal tract. In contrast, the community structure of the LF group improved, similar to that of the NC group. Hu et al. found that tacrolimus-induced changes in the intestinal flora may affect immune function and organ transplantation, and their investigation demonstrated that *Bacteroidetes* increased following tacrolimus treatment, whereas *Firmicutes* and *Proteobacteria* decreased. These findings support our results [30].

Furthermore, the abundance of *Lactobacillus* in the intestinal tract of mice treated with LF increased, which could establish a biological barrier for the intestinal mucosa, thereby improving the ability of the intestinal mucosal barrier and the immunity of the host (Figure 6D). Similarly, as depicted in Figure 6E, the bubble chart of the species distribution at the phylum level exhibited the same results as Figure 6D. The linear discriminant analysis effect size (LEfSe) was further used to analyze the effect of the treatment group on the intestinal flora of the immunosuppression mice, and the results are depicted in Figure 6F. In the MC group, the abundance of two pathogenic bacteria (*Escherichia-Shigella* and *Proteus*) increased but did not in the co-fermentation with the multi-enzyme-coupling probiotics group. These results suggested that ginseng co-fermentation with multi-enzyme-coupling *Lactobacillus rhamnosus* might repair intestinal barrier damage by affecting gut microbiota metabolism, such as increasing probiotics and reducing pathogenic bacteria, thus improving the host immune function.

## 4. Conclusions

Overall, a novel strategy for processing ginseng by using multi-enzyme-coupling probiotics was developed by screening enzymes (cellulase, compound pectinase X, and flavor protease) and probiotics (*Bacillus amyloliquefaciens* and *Lactobacillus rhamnosus*) for co-fermentation. The minor ginsenosides and polysaccharides in the ginseng fermentation broth significantly increased after co-fermentation. The lyophilized ginseng fermentation broth was used to treat the immunosuppressed mice. The results indicate that the ginseng fermentation broth treated with multi-enzyme-coupling probiotics could effectively alleviate immunosuppression and regulate intestinal flora homeostasis in mice. In conclusion, co-fermentation using multi-enzyme-coupling probiotics provided a novel strategy for the effective utilization of ginseng.

## Figures and Tables

**Figure 1 nutrients-15-02434-f001:**
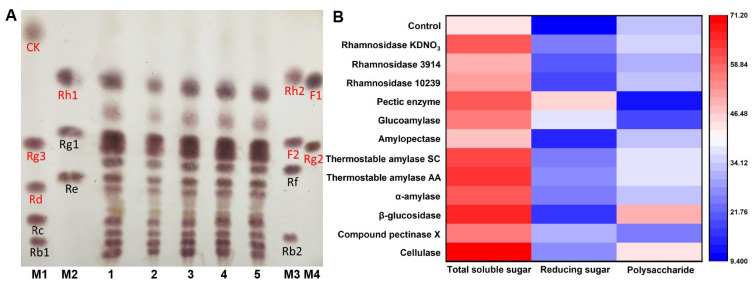
(**A**) TLC analysis of ginsenosides in ginseng fermented by different strains. Red markers represent minor ginsenosides, such as CK, Rg3, Rd, Rh1, Rh2, F2, F1, and Rg2; black markers represent major ginsenosides, such as Rc, Rb1, Rg1, Re, Rf, and Rb2 (1: Control; 2: Ginseng fermented with *Lactobacillus rhamnosus*; 3: Ginseng fermented with *Lactobacillus plantarum*; 4: Ginseng fermented with *Streptococcus thermophilus*; 5: Ginseng fermented with *Bacillus amyloliquefaciens*; M1–M4: Ginsenoside standard sample). (**B**) The total soluble sugar and reducing sugar contents in ginseng after enzymatic transformation (mg/mL).

**Figure 2 nutrients-15-02434-f002:**
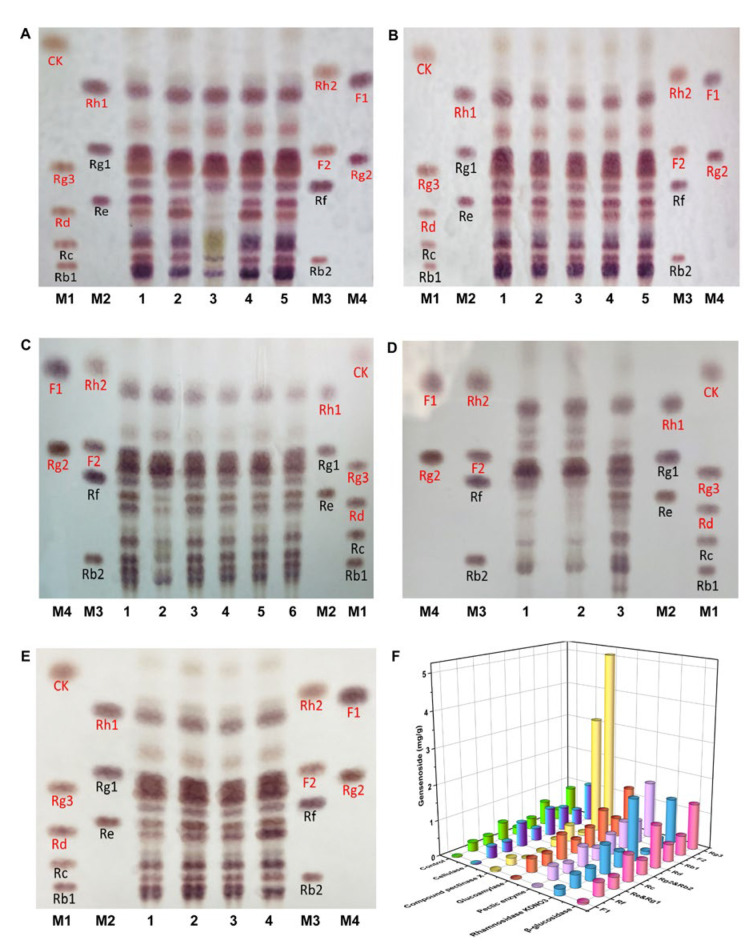
(**A**–**E**) TLC analysis of ginsenosides in ginseng fermented by different enzymes. (**A**) 1: Control; 2: Cellulase; 3: Compound pectinase X; 4: β-glucosidase; 5: α-amylase; M1-M4: Ginsenoside standard sample. (**B**) 1: Control; 2: Papain; 3: Bromelain; 4: Flavor protease; 5: ficin; M1-M4: Ginsenoside standard sample. (**C**) 1 and 6: Control; 2: Glucoamylase; 3: Pullulanase; 4: Thermophilic amylase SC; 5: Thermophilic amylase AA; M1-M4: Ginsenoside standard sample. (**D**) 1: Pectinase; 2: compound pectinase X; 3: Control; M1-M4: Ginsenoside standard sample. (**E**) 1: Control; 2: Rhamnosidase 10239; 3: Rhamnosidase 3914; 4: Rhamnosidase KDNO_3_; M1-M4: Ginsenoside standard sample. (**F**) The content of ginsenosides in the ginseng after enzymatic transformation were determined via HPLC (minor ginsenosides: Rg3, Rd, F2, F1, and Rg2; major ginsenosides: Rc, Rb1, Rg1, Re, Rf, and Rb2).

**Figure 3 nutrients-15-02434-f003:**
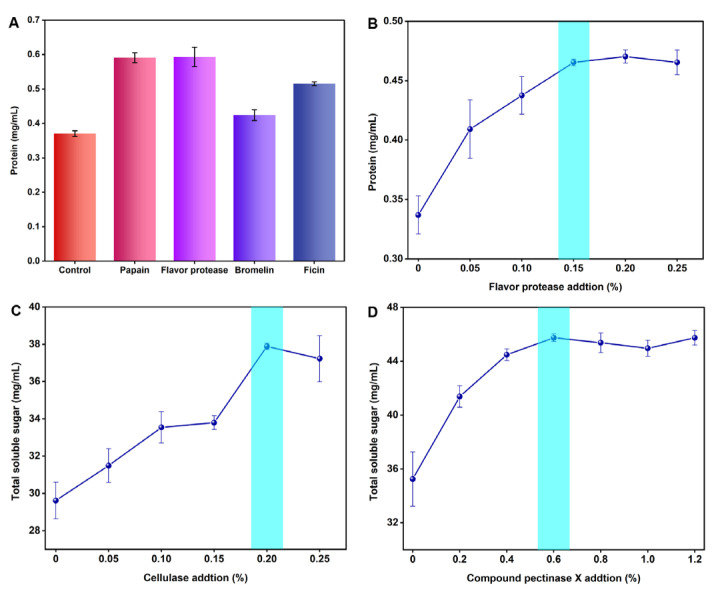
(**A**) Protein contents of ginseng after enzymatic transformation. (**B**) Effect of flavor protease addition on protein contents in ginseng. (**C**) Effect of cellulase addition on total soluble sugar content in ginseng. (**D**) Effect of compound pectinase X addition on total soluble sugar contents in ginseng.

**Figure 4 nutrients-15-02434-f004:**
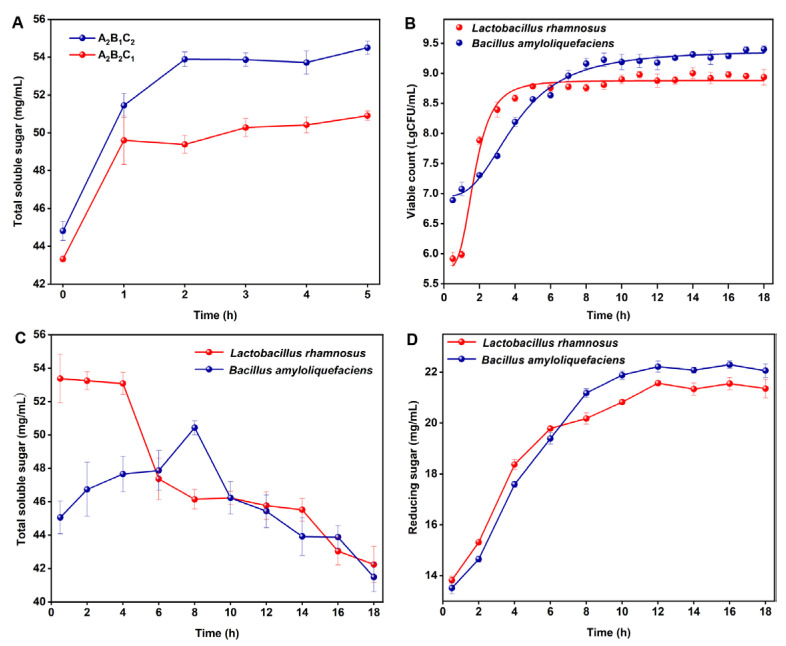
(**A**) Effect of two multi-enzyme combinations on total soluble sugar contents. (**B**) Determination of *Lactobacillus rhamnosus* and *Bacillus amyloliquefaciens* count in the fermentation process. (**C**) Determination of total soluble sugar contents during the fermentation of *Lactobacillus rhamnosus* and *Bacillus amyloliquefaciens*. (**D**) Determination of reducing sugar contents during the fermentation of *Lactobacillus rhamnosus* and *Bacillus amyloliquefaciens*.

**Figure 5 nutrients-15-02434-f005:**
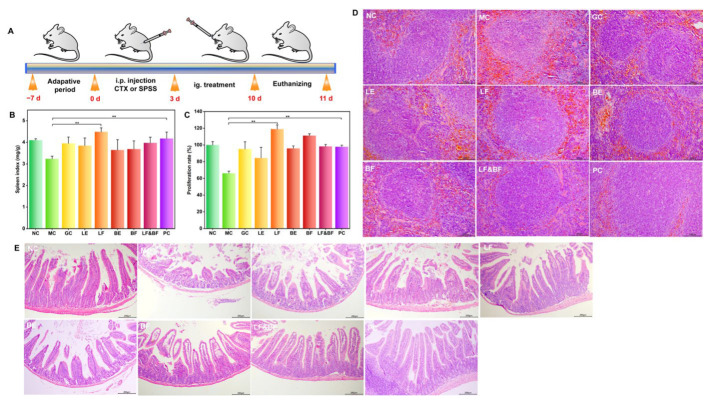
In vivo therapeutic efficacy of multi-enzyme-coupling probiotic in CTX-induced immunosuppression mice. (**A**) Experimental schedule of multi-enzyme-coupling probiotic treating CTX-induced immunosuppression mice. (**B**) Effects of different ginseng treatments on spleen index in mice. (**C**) Effects of different ginseng treatment groups on the proliferation of lymphocytes in mice. ** *p* < 0.01. (**D**) H&E staining of spleen tissue (200×). (**E**) H&E staining of intestinal tissue (100×).

**Figure 6 nutrients-15-02434-f006:**
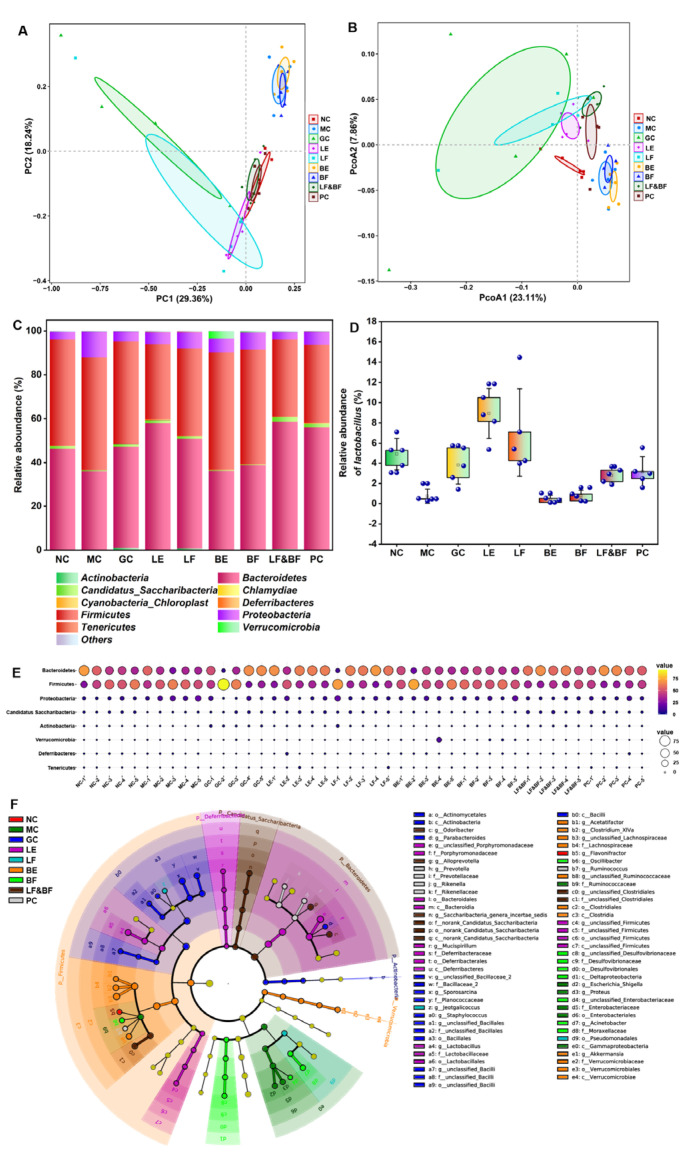
(**A**) Principal component analysis (PCA) of beta diversity of intestinal flora. (**B**) Principal co-ordinates analysis (PCoA) of beta diversity of intestinal flora. (**C**) Histogram of relative abundance of dominant species at the phylum level. (**D**) Effects of different ginseng samples on the relative abundance of *Lactobacillus* in the gut. (**E**) Bubble chart of species distribution at the phylum level. (**F**) LEfSe analysis at the genus level.

**Table 1 nutrients-15-02434-t001:** Orthogonal test design and the colony numbers for *Lactobacillus rhamnosus*.

No.	Cellulase (A)	Compound Pectinase X (B)	Flavor Protease (C)	*Lactobacillus rhamnosus* (CFU/mL)
1	1	1	1	9.1 × 10^7^
2	1	2	2	8.2 × 10^7^
3	1	3	3	5.5 × 10^7^
4	2	1	2	1.1 × 10^8^
5	2	2	3	8.3 × 10^7^
6	2	3	1	7.4 × 10^7^
7	3	1	3	1.3 × 10^7^
8	3	2	1	1.1 × 10^7^
9	3	3	2	9.6 × 10^6^
K1	7.9 × 10^7^	7.2 × 10^7^	5.9 × 10^7^	
K2	9.0 × 10^7^	5.9 × 10^7^	6.8 × 10^7^	
K3	1.1 × 10^7^	4.6 × 10^7^	5.1 × 10^7^	
R	7.8 × 10^7^	2.6 × 10^7^	1.7 × 10^7^	
optimal level	A_2_	B_1_	C_2_	
major factor	A > B > C			
optimal formula	A_2_B_1_C_2_			

**Table 2 nutrients-15-02434-t002:** Orthogonal test design and the colony numbers for *Bacillus amyloliquefaciens*.

No.	Cellulase (A)	Compound Pectinase X (B)	Flavor Protease (C)	*Bacillus amyloliquefaciens* (CFU/mL)
1	1	1	1	2.7 × 10^6^
2	1	2	2	1.1 × 10^8^
3	1	3	3	4.3 × 10^7^
4	2	1	2	3.8 × 10^6^
5	2	2	3	1.4 × 10^8^
6	2	3	1	5.3 × 10^7^
7	3	1	3	1.5 × 10^6^
8	3	2	1	1.4 × 10^8^
9	3	3	2	3.9 × 10^7^
K1	5.1 × 10^7^	2.6 × 10^6^	6.5 × 10^7^	
K2	6.5 × 10^7^	1.3 × 10^8^	5.0 × 10^7^	
K3	6.0 × 10^7^	4.5 × 10^7^	6.1 × 10^7^	
R	1.4 × 10^7^	1.3 × 10^8^	1.5 × 10^7^	
optimal level	A_2_	B_2_	C_1_	
major factor	B > C > A			
optimal formula	A_2_B_2_C_1_			

## Data Availability

No new data were created or analyzed in this study. Data sharing is not applicable to this article.

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
