# Peer review of "Ginsenosides and Polysaccharides from Ginseng Co-Fermented with Multi-Enzyme-Coupling Probiotics Improve In Vivo Immunomodulatory Effects"

_nutrients, 2023, doi:10.3390/nu15112434_

Round 1
Reviewer 1 Report
Interesting paper on ginsenoside and ginseng polysaccharide co-fermented with multienzyme-coupled probiotics improves immunomodulatory effects in vivo.
Appropriate methodology for the defined objectives and
appropriate statistical analysis of the collected data.
Discussion and conclusions based on the results obtained in mice.
Author Response
First, we would like to thank the reviewer for the critical examination of our manuscript. The comments are very helpful for revising the manuscript. The attached cover letter and manuscript have been carefully revised and point-by-point response.

Reviewer 2 Report
Write down the correct name of the plant.
Specify the place of harvesting the plant by giving geographical coordinates. Description of working methods for TLC and HPLC.
What kind of standards were used for HPLC. For the Bradfort method, the description and results are missing. Explain why you used Cyclophosphamide in the experimental model used.
Author Response

(The authors gave the same response as above.)

Reviewer 3 Report
In the manuscript „ Ginsenoside and polysaccharide from co-fermented ginseng with multi-enzyme coupling probiotics improves the immunomodulatory effects in vivo.“ authors present their work on co-fermenting ginseng with several enzymes and two probiotic strains (Lactobacillus rhamnosus and Bacillus amylolyticus ) in order to obtain ginseng fermentation broth with higher ginsenoside, polysaccharide, protein and probiotics level. The immunomodulatory effects of prepared products were tested in vivo using immunosuppressive mice models. The work is interesting, especially in aiming to increase the amount of ginseng active compounds obtained in product. However, some improvements, mainly regarding the presentation of results, must be made and my suggestions are in the attached file.

Author Response

(The authors gave the same response as above.)

Round 2
Reviewer 2 Report
Accepted for the publication
Reviewer 3 Report
The authors have successfully addressed all issues from the reviewing process. I believe that provided additions have improved the manuscript and it is now suitable for publication.